# Exploring High-Resolution Chemical Distribution Maps of Incompatible and Scarce Metals in a Nepheline Syenite from the Massif of "Serra de Monchique" (Portugal, Iberian Peninsula)

Sofia Barbosa [1,*], António Dias [2], Diogo Durão [3], José Grilo [3], Gonçalo Baptista [3], Jonhsman Cagiza [1], Sofia Pessanha [2], Joaquim Simão [1] and José Almeida [1]

[1]   GeoBioTec—GeoBioSciences, GeoTechnologies and GeoEngineering & NOVA FCT, 2829-516 Caparica, Portugal
[2]   LIBPhys & NOVA FCT, 2829-516 Caparica, Portugal
[3]   NOVA FCT, 2829-516 Caparica, Portugal
*   Correspondence: svtb@fct.unl.pt

**Abstract:** In this case study, 2D micro energy dispersive X-ray fluorescence (μ-EDXRF) surveys were performed in the nepheline syenite (NS) of "Serra de Monchique" located in the southwest region of Portugal (Algarve, Iberian Peninsula). The results allow the identification in the mineral matrix of certain elements classified as critical raw materials (CRMs). Due to substitution effects, some scarce transition elements, such as Zn and Ni, are present and camouflaged in alkali silicate minerals, while others, such as Co, are included in ferromagnesian mineral phases. As expected, incompatible elements are preferably distributed on the surface of aluminosilicate mineral phases such as Rb and Ga, or exclusively in K-bearing feldspar phases, as it is the case of Sr. Interesting CRMs such as Ti, Zr, and Nb are well individualized in oxides, as well as in sphene and apatite. The detected antagonistic chemical distribution between Ti and Fe, and the good spatial relation between Ti and Ca confirms that Ti is present as sphene and, in areas with absent Si, probably occurs as rutile. Nb has a distribution pattern quite similar to Zr and occurs due to substitution effects. It was possible to conclude that there is probable co-existence of Zr-REE-Nb in specific mineral phases such as apatite, zircon, and other Zr-oxides. These results evidence and confirm NS as a potential source of multiple industrial minerals and distinct scarce elements which are incorporated in oxide or phosphate phases that can be more effectively separated in the beneficiation process.

**Keywords:** critical raw materials; industrial minerals; micro X-ray elemental mapping; micro X-ray image co-localization; RGB clustering image analysis; Zr-REE-Nb

## 1. Introduction

The growing global demand for alternative sources of distinct types of strategic industrial minerals and scarce metals evidences the inevitable need and obligation on the part of each nation or administrative region to increase its respective degree of knowledge in relation to its geological resource base. The present study was developed in this context and its aim was to identify distinct potential critical raw materials (CRMs) in nepheline syenites of the "Serra de Monchique", located in the northwest region of the Algarve region in southern Portugal (Iberian Peninsula). Nepheline syenites (NS) are well known for their content of distinct elements classified as critical raw materials (CRMs). It is known that original alkaline magmas are greatly enriched in many predominantly incompatible large ion-lithophile elements (LILEs) and high-field-strength elements (HFSEs), and that rare metals that can be present in distinct types of NS. Niobium (Nb), beryllium (Be), caesium (Cs), lithium (Li), tantalum (Ta), zirconium (Zr), yttrium (Y), and rare earth elements (REE), thorium (Th), uranium (U), and hafnium (Hf) as well as iron (Fe), titanium (Ti), zinc

(Zn), copper (Cu), vanadium (V), chromium (Cr), fluor (F), phosphor (P), chlorine (Cl), among others, are typically identified in elemental composition of nepheline syenite [1–4]. Niobium (Nb) is present in highest concentrations in alkaline rocks as NS, and in its weathering products such as bauxite deposits [5]. In general, alkaline granite and syenite complexes display outstanding associations of rare minerals incorporating the HFSEs as major elements such as Zr, Nb, Y, U, and distinct REE [4,6]. A substantial part of these commodities (i.e., REE, Y, and Nb) has a critical character indicative of supply risk for political or environmental reasons being classified as a CRM [4].

In general terms, NS is a brownish gray light-colored medium- to coarse-grained complex rock consisting of different mineral phases [7]. It is a plutonic igneous rock largely made up of alkali-feldspar (orthoclase, microcline, $KAlSi_3O_8$), nepheline (Na, K) $AlSiO_4$, few sodium feldspars (albite, $NaAlSi_3O_8$), and without any free silica [7]. Nepheline syenites are distinguished from ordinary basic syenites not only by the presence of nepheline but also by the occurrence of many other minerals rich in alkalis, rare earth elements, and incompatible elements [5]. More petrographic research has been devoted to nepheline syenites than to any other plutonic rock due to their extraordinarily varied mineralogy and their remarkable variation in habit, fabric, appearance, and composition [5]. Nepheline is sometimes wholly or partly replaced by sodalite $Na_4Al_3(SiO_4)_3Cl$, the principal feldspathoid mineral, in addition to nepheline, or cancrinite, a complex feldspathoid mineral of carbonate, and silicate of sodium, calcium, and aluminium, $Na_6Ca_2[(CO_3)_2 | Al_6Si_6O_{24}]\cdot 2H_2O$, which occurs in several nepheline-syenites. Other minerals that are common in minor amounts include sodium-rich pyroxene (aegirine-augite), biotite, titanite, iron oxides, apatite, fluorite, melanite garnet, and zircon [5]. The commonest dark silicate is green pyroxene. In some areas, pyroxene is virtually absent, and it is replaced by a mixture of hornblende and biotite. Alkaline amphibole is also abundant. Extremely iron-rich olivine is rare but may be present in some nepheline syenite [5].

Nepheline and feldspar are highly sought industrial minerals for ceramics, glass manufacture, plastics, cosmetics, paint, and building products [8,9]. When added to ceramic and glass products, nepheline lowers the melting point of silica reducing the amount of energy required for glass production as well as increasing hardness and durability [8,10]. Nepheline is also favored as a paint additive as it is silica-free, and in Russia is used as a source for aluminium [8,10]. However, some inclusions within the felspathic and feldspathoid minerals, as well as accessory minerals present in the rock matrix affect extremely negatively the desired values for this raw material in the ceramic-glass industries [11]. In fact, impurities present in feldspathic minerals lead to quality problems [9,11]. These impurities generally result from the presence of inclusion minerals such as hornblende, biotite, chlorite, apatite [12,13], and sphene. Iron, titanium, mica, and calcite minerals in NS can lead to quality problems on the surface of floor tiles because of different sintering properties. This problem increases with weathering effects [11]. Impurities in feldspathic minerals are generally determined by examining the content of element values such as Fe, Ti, Mn, or Mg [9,11]. In this context, it is relevant to ensure the elimination of this impurities from nepheline syenite prior to the utilization of its feldspathic minerals in industry.

In this exploratory study, 2D micrometric map surveys were performed in a sample from the nepheline syenite of "Serra de Monchique" (Algarve, Portugal). As some of the most relevant CRMs in nepheline syenite occur in fine to very fine particle grain size fractions, application of micro–X-ray fluorescence (μ-XRF) map surveys 2D were tested as an elemental high resolution investigation technique. 2D high-resolution element mapping through μ-XRF of mineral rock matrix is an emergent and challenging area of investigation. In spectroscopy, μ-XRF computed tomography systems help in understanding the factors that control critical processes at distinct microscopic scales [14]. It can provide increasingly reliable information at ~25 μm resolutions. μ-XRF is a non-destructive technique that leaves samples intact for other types of analyses, such as Raman spectroscopy, which allows characterization of molecular components [14,15]. In [15,16] the authors

evidence the suitability of this technique for various applications within the earth sciences. 2D high-resolution chemical distribution maps can be used as qualitative multi-element maps, as semiquantitative single-element maps, and as a basis for a novel image analysis workflow quantifying the modal abundance, size, shape, and degree of sorting of segmented components [16]. They rapidly produce bulk and phase-specific geochemical data sets [16]. Elemental quantifications, 2D, and even 3D mapping surveys at microscale can be performed [14,15].

## 2. Materials and Methods

### 2.1. The Nepheline Syenite of "Serra de Monchique" Massif

The Monchique massif is located in Southwest Portugal, in the Algarve region, dating from the Upper Cretaceous, between $76 \pm 4$ Ma [17–21]. It is a sub-volcanic laccolith that intrudes Paleozoic metasediments (Figure 1) [19–23]. This massif can be included in the Iberian Alkaline Igneous Province, which encompasses several intrusions dating from the Upper Cretaceous, being contemporary with the Sines and Sintra massifs and integrating with them an important geological alignment (Figure 1) of the Iberian Alkaline Igneous Province [24]. NS intrusions can be subdivided according with the agpaitic index $(Na + K)/Al$ or $(Na_2O + K_2O)/Al_2O_3$, being less alkaline (miaskitic group) or more alkaline (agpaitic group) [5]. Much of the Monchique massif is made up of nepheline syenite ([19,23], (Figure 1). Its body has a zoned structure with semi-elliptical shape, outcropping in an area of 80 km$^2$. It is the most important alkaline igneous massif in Europe and one of the largest massifs of miaskitic composition syenites in the world [25–28]. The strong lineaments that cut the massif do not displace the internal and/or external contacts existing cartographic continuity between the units [23]. The cartography works carried out by [23,29] confirmed the multi-compositional character of the complex and its concentric zoned structure. The cartographic review of these authors allowed the individualization of two distinct facies of syenite and the identification of a typical cartographic pattern of zoned intrusions. They defined a concentric structure composed of two distinct units: (a) an homogeneous central unit, composed of medium to coarse grained NS and (b) an edge unit, mineralogically and texturally more heterogeneous, formed by syenites, of finer granularity with varying proportions of nepheline. In the contacts between the two predominant facies, or inside the onboard unit, but close to the contact with the central unit, these authors recorded the occurrence of several bodies of ultramafic, mafic, and intermediate rocks and also breccia formations.

Macroscopically, the NS from Monchique is greyish-brown leucocratic rock, with a medium to coarse grained phaneritic texture being possible to visualize the elongated well-developed habit of K-feldspars and reddish-brown nepheline grains [10,28] (Figure 1). The nepheline syenite of Monchique is mainly made up of K-feldspar, followed by nepheline and, in smaller amounts, aegirine-augite, sphene, and biotite [10,28]. Under the petrographic microscope, K-feldspar is often very altered and presents Carlsbad-twinned crystals. Nepheline grains tends to be little and altered, aegirine-augite is fractured, sphene is observed in diamond shape euhedral crystals, and biotite in small crystals. Opaque minerals are also observed. In [28] the percentages obtained of essential minerals for this NS were: 45% K-feldspar, 22% nepheline, 10% aegirine-augite, 8% sphene, and 6% biotite. The remaining percentage relates to accessory minerals, namely: sodalite, hornblende, apatite, rutile, zircon, and opaque minerals.

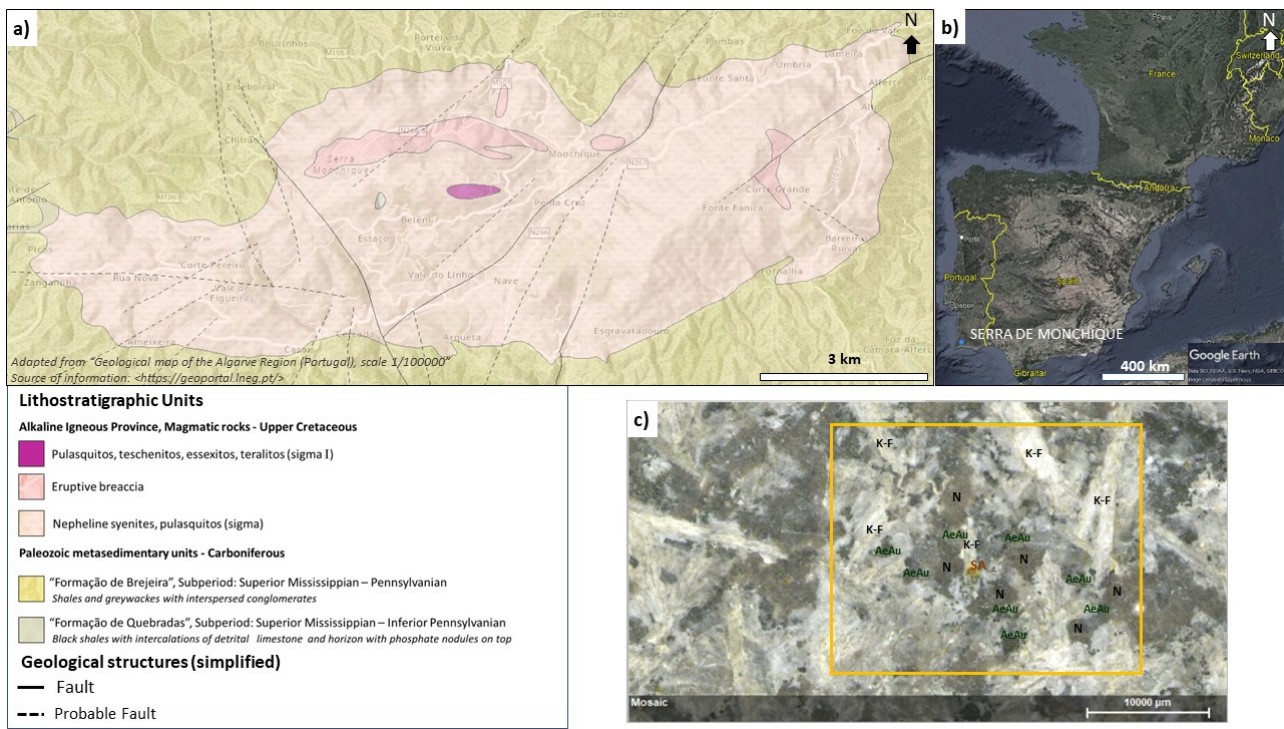

**Figure 1.** (**a**) Lithostratigraphic units in Serra the Monchique massif and its surroundings and main geological structures (adapted from "Geological Map of the Algarve Region (Portugal), scale 1/100000", source of information: https://geoportal.lneg.pt/ (accessed on 25 July 2022)); (**b**) Localization of the "Serra de Monchique" massif in the context of southwestern Europe and Iberia; and (**c**) tested sample of the "Serra the Monchique" nepheline syenite (polished surface with indication of the area selected for μ- EDXRF surveys) with the indication of some of the occurrences of mineral phases nepheline (N), K-felspar (K-F), and aegirine-augite (Ae-Au). Mineral superficial alteration (SA) is also indicated.

In moderately peralkaline miaskitic NS, Ti, Zr, and other HFSEs are hosted in common minerals such as ilmenite, titanite, and zircon [5]. Studies developed by [24] indicate significant concentrations of Zr and REE lanthanide (La) in 34 samples collected in the "Serra de Monchique" alkaline massif and analyzed for major, and trace, elements and REEs by inductively coupled plasma (ICP), instrumental neutron activation analysis (INAA) and X-ray fluorescence. Zr presented concentrations generally above 200 parts per million (ppm) having reached 800 ppm to 1400 ppm, and for La concentrations were generally above 60 ppm reaching the maximum of 160 ppm [24]. Also, in [29] the author concluded that the variations of elemental concentrations were as follows: Rb (28 to 214 ppm), Ba (165 to 1230 ppm), Sr (291–3280), Cs (0.5 to 4.8 ppm), V (33 to 462 ppm), Sc (2 to 29 ppm), Co (2 to 49 ppm), Y (10 to54 ppm), Nb (78 to 168 ppm), total light REEs (160 to 836 ppm), and total heavy REEs (6 to 51 ppm).

## 2.2. Micro X-ray Fluorescence Elemental Mapping

The micro X-ray fluorescence technique was applied by means of the energy dispersive X-ray spectrometer M4 TORNADO by Bruker (Billerica, MA, USA). This instrument consists of a low-power X-ray tube with a Rh anode and was operated at 50 kV and 300 μA. Placed after the X-ray tube, a poly-capillary lens focuses the beam to a spot size that can go down to 25 μm for X-ray mass absorption coefficients (Mo-Kα). This way, by selecting an area in the sample, point-by-point measurements are performed to identify the existent elements and elemental 2D images surveys are generated.

In the case study, a rock sample with a polished surface was analyzed with Al 12.5 μm and AlTiCu 100/50/25 μm filter composition (that is., thickness of Al 100 μm, Ti 50 μm and Cu 25 μm). For elements emitting radiations from 5 to 35 keV it is adequate to use filters

that can lessen the effect of the Bremsstrahlung radiation that contribute to background radiation [30]. Therefore, for NS samples the two filters mentioned above were used due to the presence of elements with atomic numbers (Z) superior to 21, i.e., from titanium (Ti) to yttrium (Y). One of the advantages of µ-EDXRF 2D is that it can be applied to a surface with different polishings (coarse to softened). The polishing procedure was adopted considering the usual and applicable best industrial practices techniques. The sample was first cut using a diamond saw, and afterwards a course to fine grinding of the cut surface was done using silicon carbide (SiC) abrasive powders. Ultrasonic cleaning was applied in the lab after the cutting and after the course and fine grinding and before the measurements to ensure that no other material would remain on the surface.

The measurements were done under 20 mbar vacuum conditions (to improve detection limits), with a step size of 15 µm and 10 ms acquisition per spectrum rendering, in an average of one hour and thirty minutes to ensure high resolution 2D maps for each element.

Data treatment of micro-2D mapping was performed using the inbuilt software MQuant.

### 2.3. 2D Image Mapping Processing: Clustering RGB Pixel Analysis

Micro-EDXRF 2D mapping outputs consisted of 2D image files. Possibilities related to the processing of these image files are dependent on pixel quantification and statistical analysis of its distributions. In this case-study each image refers to a certain element for which its occurrence and concentration are locally represented by a certain intensity of a RGB (red, green, blue) color. The highest elemental concentrations are represented by more RGB light color proportions.

Pixel proportion quantifications per distinct RGB color and its respective intensities, that is, per distinct chemical elements were established in R©v.4.0.3 2020 The R Foundation for Statistical Computing Platform, Vienna, Austria [31–34] with R©Countcolors Package v.0.9.1 Brown University Repository, Rhode Island, USA [35,36]. This package is the result of a collaboration between Sarah Hooper, Sybill Amelon, and Hannah Weller, and it was developed originally with the aim of quantifying the area of white-nose syndrome infection of bat wings. R©Countcolors Package allows users to quantify regions of an image by distinct colors. It is an R package that counts colors within specified color ranges in image files and provides a masked version of the image with targeted pixels changed to a different selected color by the user. This package integrates techniques from image processing without using any machine learning, adaptive thresholding, or object-based detection, making it reliable and easy to use although of it has limitations in more complex interpretations.

The principles and application of R©Countcolors Package to process and analyse µ-EDXRF 2D images in materials of mineral origin are described in [37,38]. In this methodology after selecting the color clusters that are the most representative for a certain element occurrence, its respective area of occurrence is estimated in percentage (%). In this case-study, and according to the methodology described in [37,38], a central color and a search radius around it, where a "sphere" for the considered color range is drawn (spherical range) was selected. This methodology was applied to perform a semi-quantitative analysis of the images of micrometric chemical occurrence of Nb and Zr.

### 2.4. Elemental Spatial Co-Localization in µ-EDXRF Maps

Image co-localization analysis of Nb and Zr was performed using R© colocr Package v.0.1.1 rOpenSci Repository, Berkeley, California, USA [39,40] which is available on the comprehensive R archive network, and the source code is available on GitHub under the GPL-3 license as part of the rOpenSci collection.

This package provides a simple straight-forward workflow for loading images, choosing regions of interest (ROIs), and calculating co-localization statistics. Included in the package, is a shiny app that can be invoked locally to interactively select the regions of interest in a semi-automatic way [39,40].

The R©colocr Package provides a straightforward workflow for determining the amount of co-localization. This workflow consists of two simple steps: (1) to choose the regions of interest (ROI) and (2) to calculate the correlation between the pixel intensities. The statistics that are used in this package to measure co-localization are Pearson's correlation coefficient are (PCC), and Manders overlap coefficient (MOC). As defined in [38], PCC is the co-variance of the pixel intensity from the two channels. The mean of the intensities is subtracted from each pixel which makes the coefficient independent of the background level. MOC is a coefficient that doesn't require subtraction of the mean. Therefore, the values are always between 0 and 1. MOC is independent from signal proportionality.

## 3. Results

### 3.1. Subsection

In the sequence of the surveys developed, it was possible to obtain 2D μ-EDXRF elemental distribution maps for silicon (Si), aluminium (Al), potassium (K), sodium (Na), iron (Fe), calcium (Ca), manganese (Mn), magnesium (Mg), titanium (Ti), barium (Ba), rubidium (Rb), strontium (Sr), zirconium (Zr), gallium (Ga), phosphorus (P), chlorine (Cl), niobium (Nb), zinc (Zn), nickel (Ni), chromium (Cr), cobalt (Co), sulphur (S), copper (Cu), and arsenic (As). In this study, interpretations are presented for the elements Si, Al, K, Na, Fe, Ca, Mn, Mg, Ti, Ba, Rb, Sr, Zr, Ga, P, Cl, Nb, Zn, Ni, and Co. For these elements, several facts or evidence can be observed, such as:

1.  Silicate and K-bearing feldspar phases are very well distinguishable in the mineral matrix through the elements Si, Al, K, Cl, and Na (Figure 2);
2.  Correspondence between Cl and P, which may indicate localization of phosphate mineral apatite, is almost absent or inexistent (Figure 3);
3.  Incompatible elements Rb, Ga, Sr, Ba, and some trace metals such as Zn and Ni accompany Si, Al, and K throughout the mineral matrix (Figures 4–7);
4.  Elements that can be considered as ferromagnesian mineral indicators, such as Fe, Ca, Mg, Mn, and Co, have a total antagonist distribution relatively to Al, and partially to Si (Figures 8 and 9);
5.  Ti has an antagonistic spatial distribution with Fe, and a very good spatial correspondence with Ca (Figure 10);
6.  Zr has a good spatial correspondence with Ca and Si (Figure 11), and partially with Ti (Figure 12);
7.  Nb and Zr have very good spatial correspondence at micrometric scale (Figure 13) in areas with and without Si;
8.  Nb, Ca, and P, have good spatial correspondences (Figures 14 and 15).

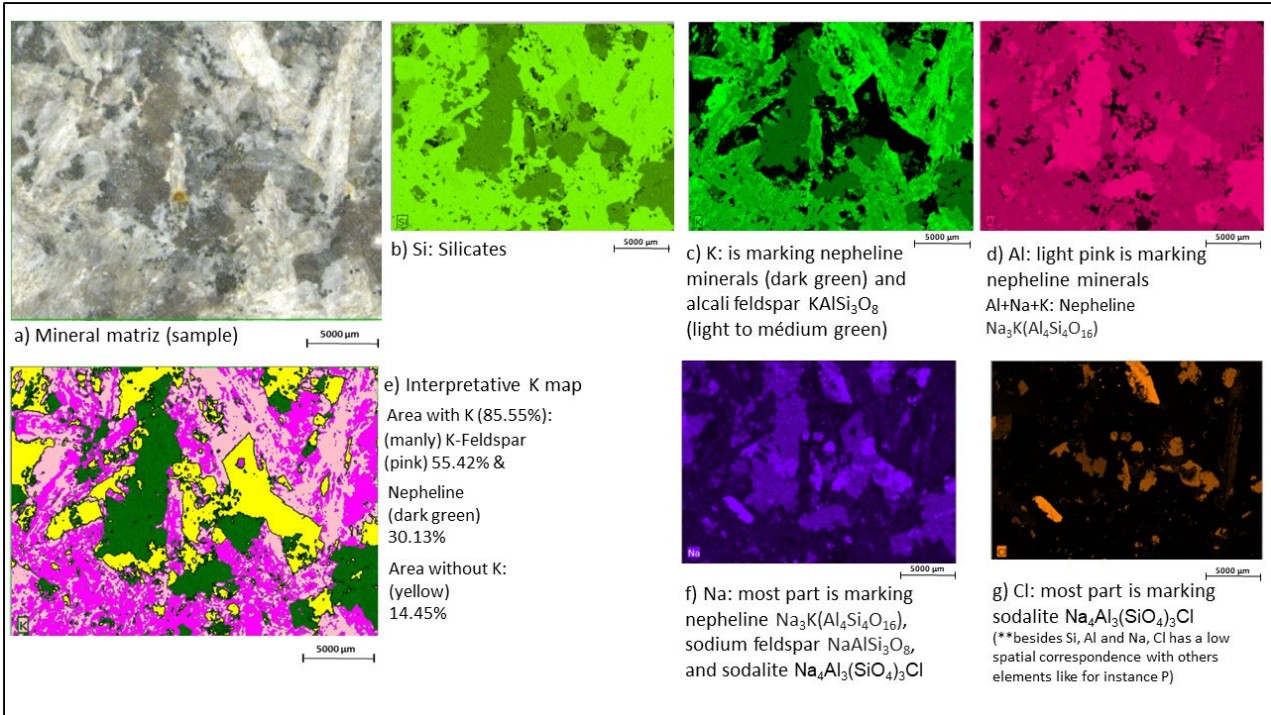

**Figure 2.** Original mineral sample, μ-EDXRF and interpretative maps: (**a**) Image of mineral matrix (sample), (**b**) Si μ-EDXRF map; (**c**) K μ-EDXRF map, (**d**) Al μ-EDXRF map, (**e**) interpretative K map, (**f**) Na μ-EDXRF map, and (**g**) Cl μ-EDXRF map.

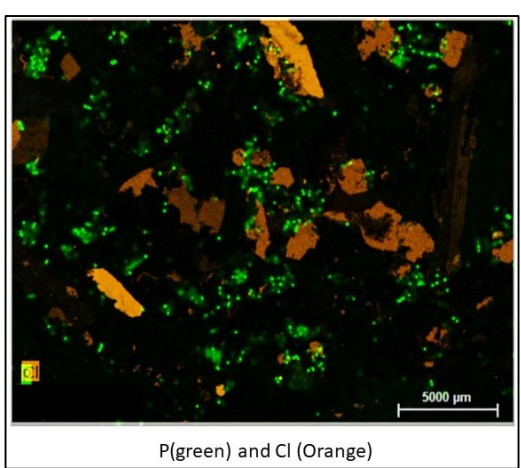

**Figure 3.** P (green) and Cl (orange) μ-EDXRF maps.

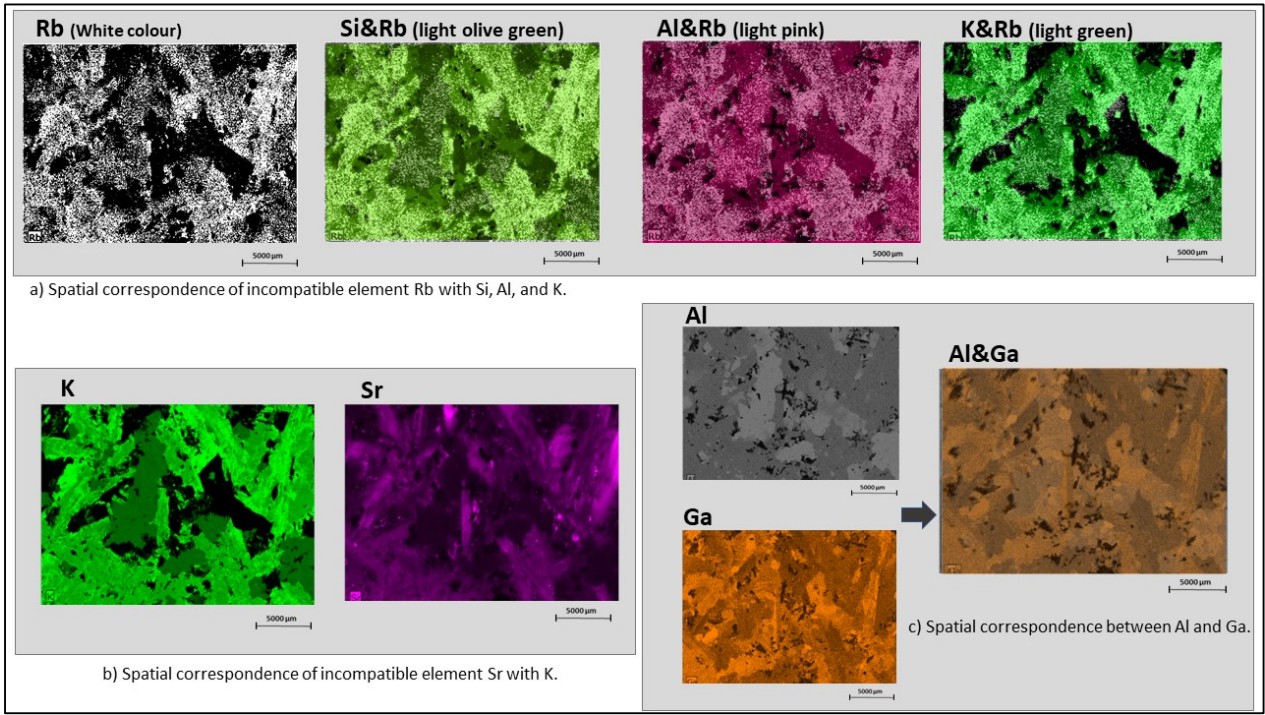

**Figure 4.** Spatial distribution and correspondence between (**a**) Rb and Si, Al, and K, (**b**) Sr and K, and (**c**) Ga and Al.

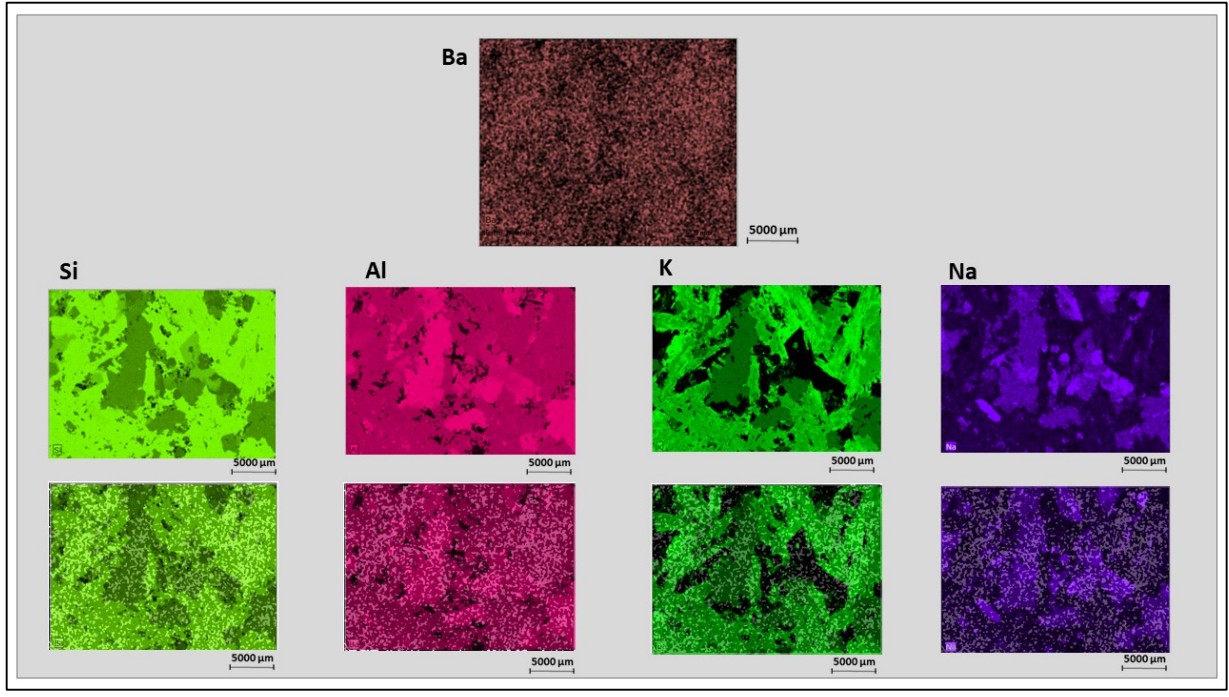

**Figure 5.** Spatial distribution and correspondence between Ba and Si, Al, K, and Na.

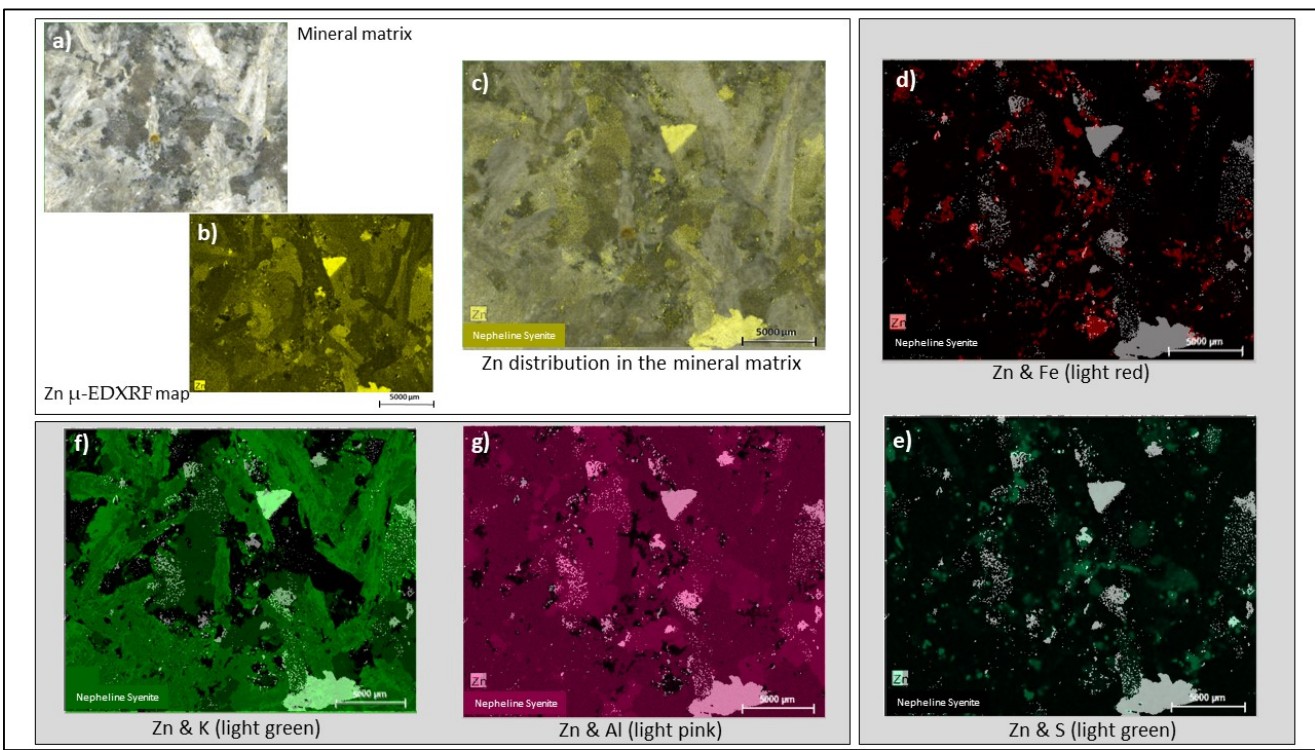

**Figure 6.** (**a**) Image of original mineral sample (mineral matrix), (**b**) Zn μ-EDXRF map, (**c**) Zn distribution in the mineral matrix (overlap of images (**a**,**b**)). Zn distribution and its correlations with (**d**) Fe, (**e**) S, (**f**) K, and (**g**) Al.

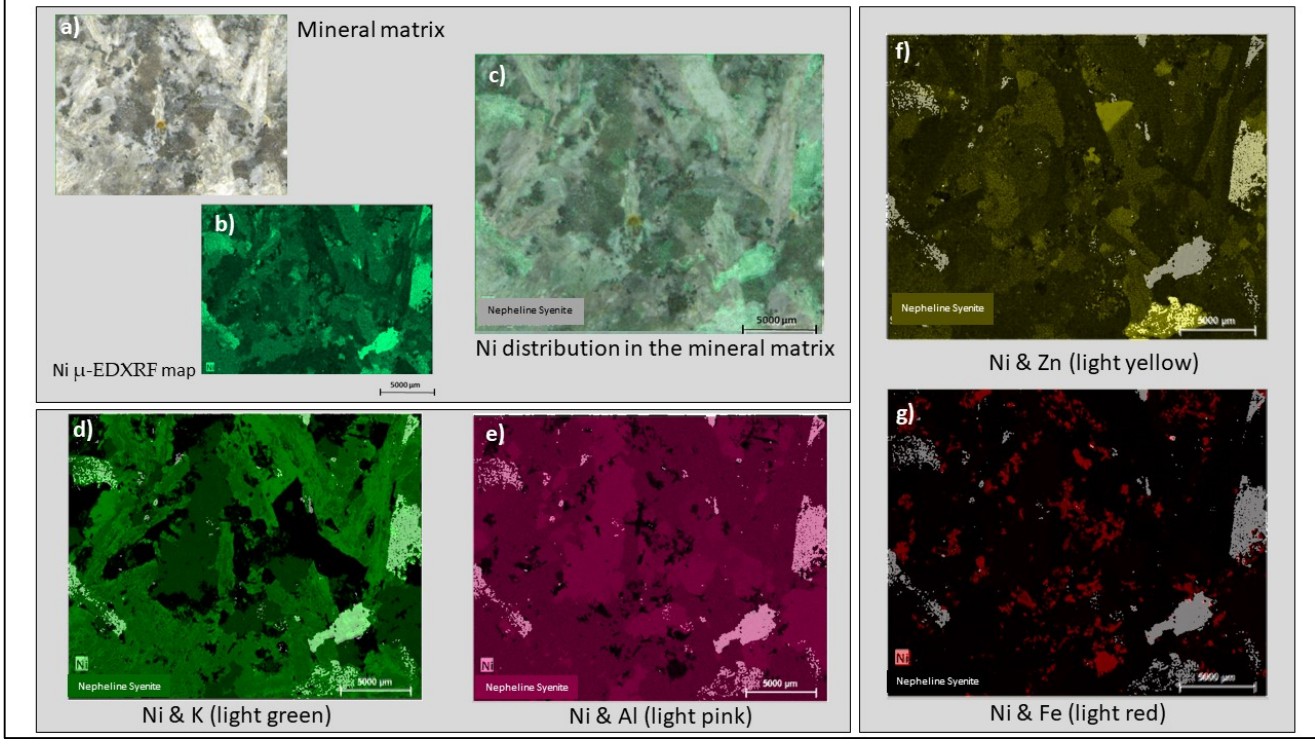

**Figure 7.** (**a**) Image of original mineral sample (mineral matrix), (**b**) Ni μ-EDXRF map, (**c**) Ni distribution in the mineral matrix (overlap of images (**a**)), and Ni distribution and its correlations with (**d**) K, (**e**) Al, (**f**) Zn, and (**g**) Fe.

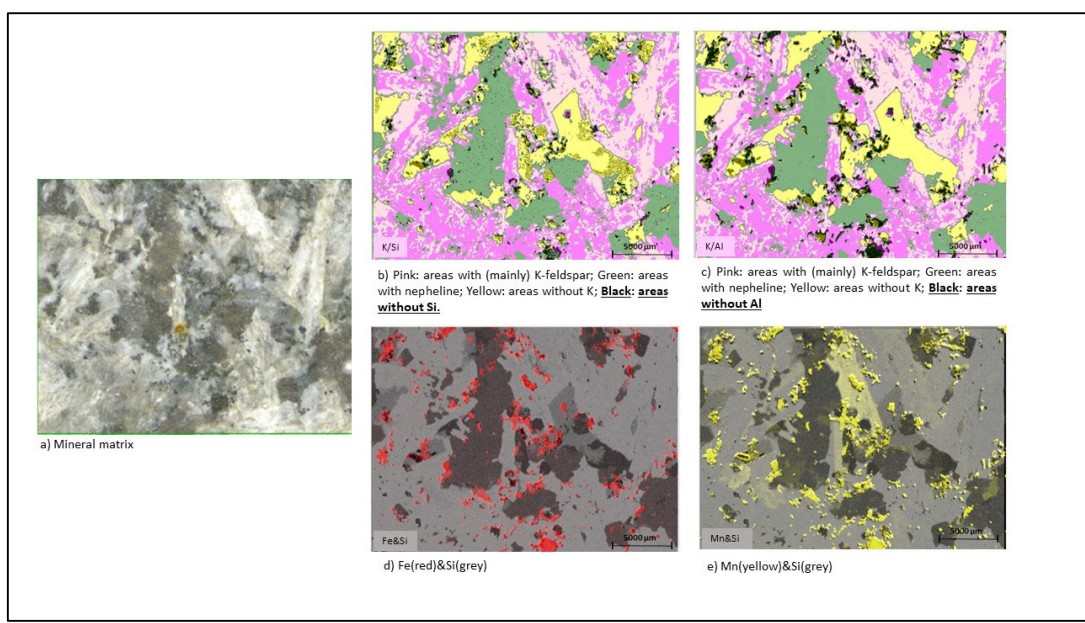

**Figure 8.** (**a**) Image of original minera sample (mineral matrix). Interpretative maps with evidence of (**b**) areas without Si and without K or (**c**) areas without Al and without K. Distributions of (**d**) Fe, and (**e**) Mn relative to Si.

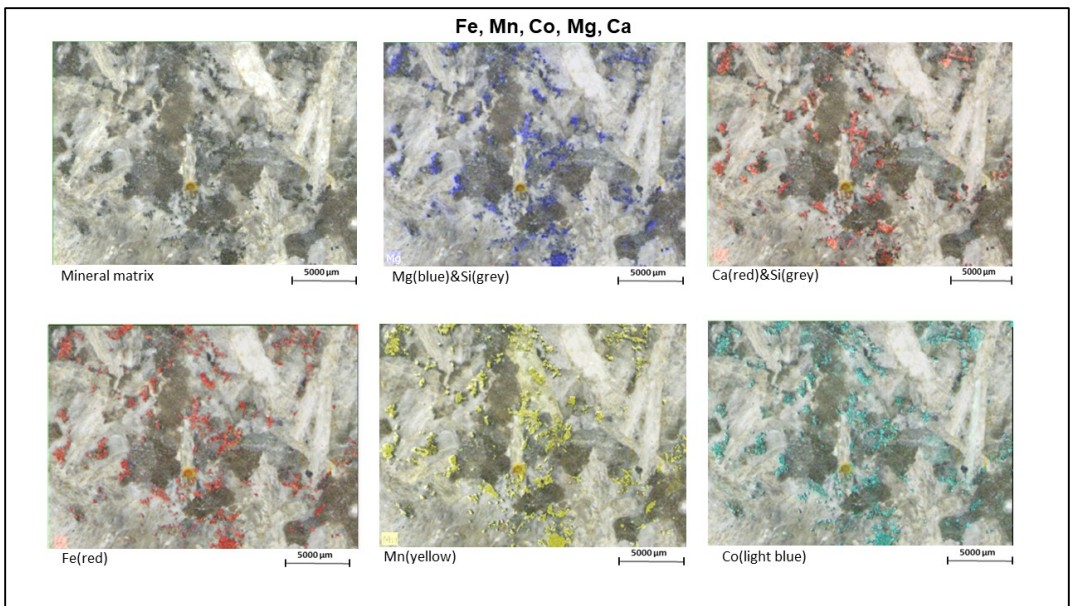

**Figure 9.** Elemental spatial micrometric distribution of Ca, Mg, Fe, Mn, and Co in the mineral matrix.

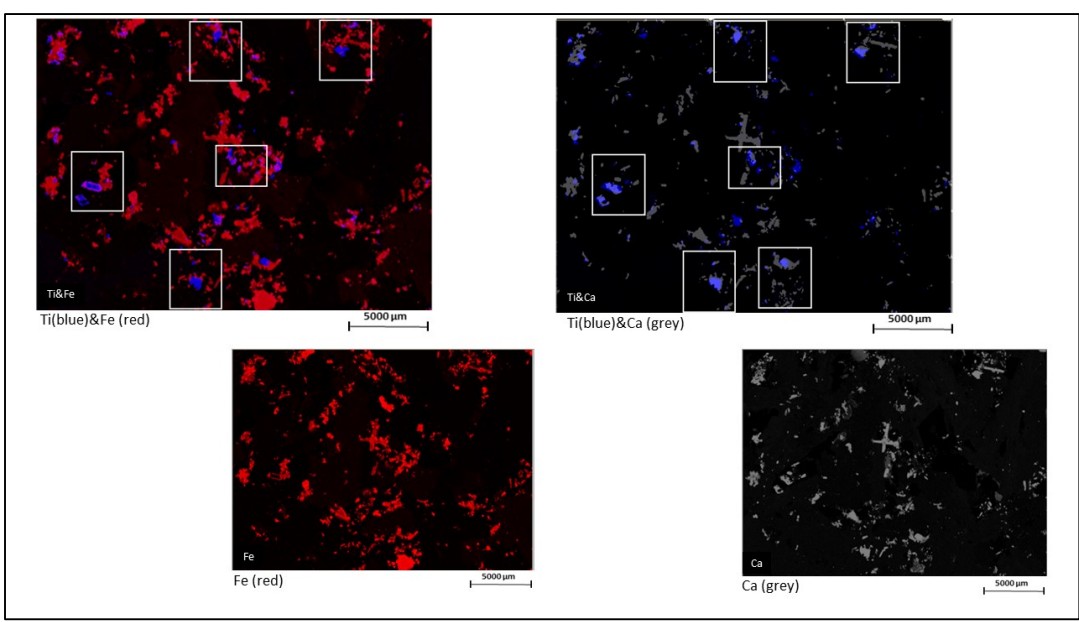

**Figure 10.** Interpretative overlap of spatial micrometric distribution maps of Ti with Fe and Ca.

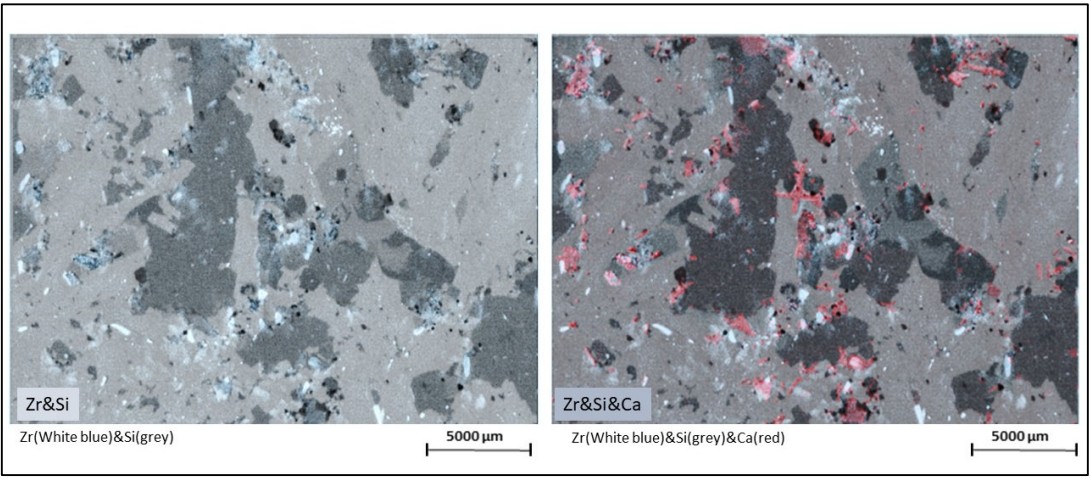

**Figure 11.** Elemental spatial micrometric distributions between Zr&Si and Zr&Si&Ca.

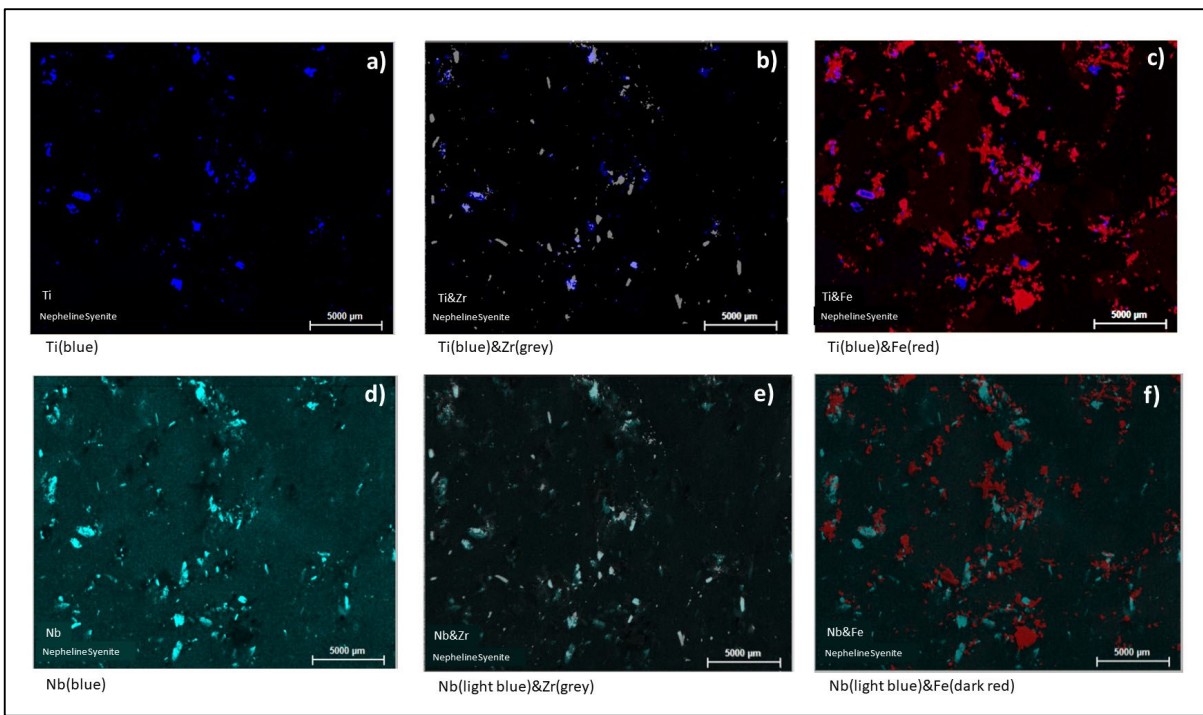

**Figure 12.** Elemental spatial micrometric distribution of (**a**) Ti, (**b**) Ti&Zr, (**c**) Ti&Fe, (**d**) Nb, (**e**) Nb&Zr, and (**f**) Nb&Fe.

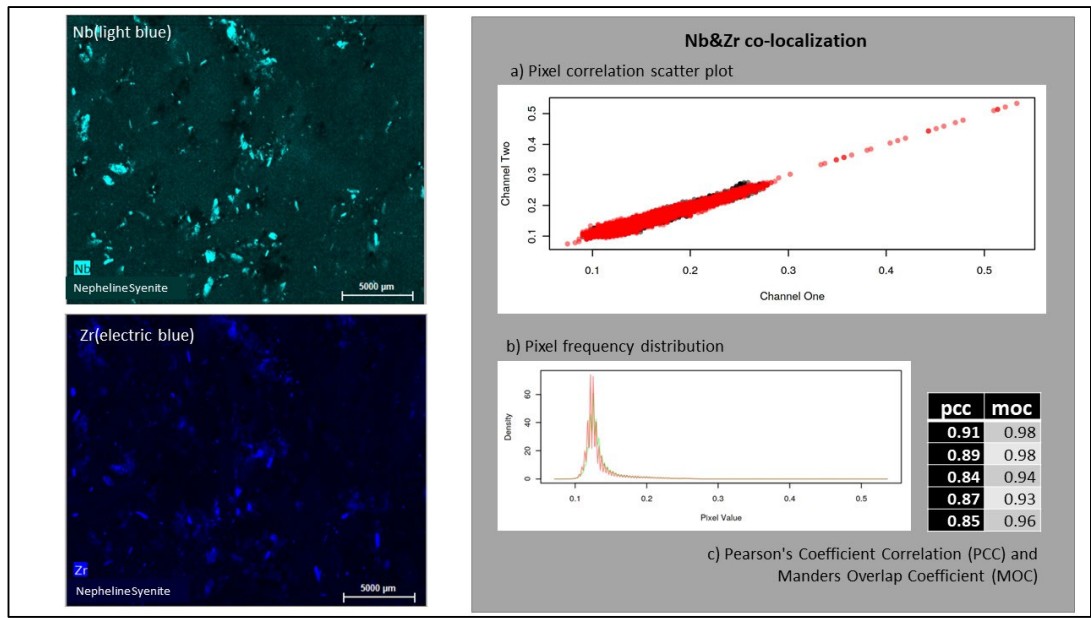

**Figure 13.** Nb and Zr μ-EDXRF maps and their co-localization statistics. (**a**) Pixel correlation scatter plot, (**b**) pixel frequency distribution, and (**c**) Pearson's coefficient correlation (PCC) and Manders overlap coefficient (MOC).

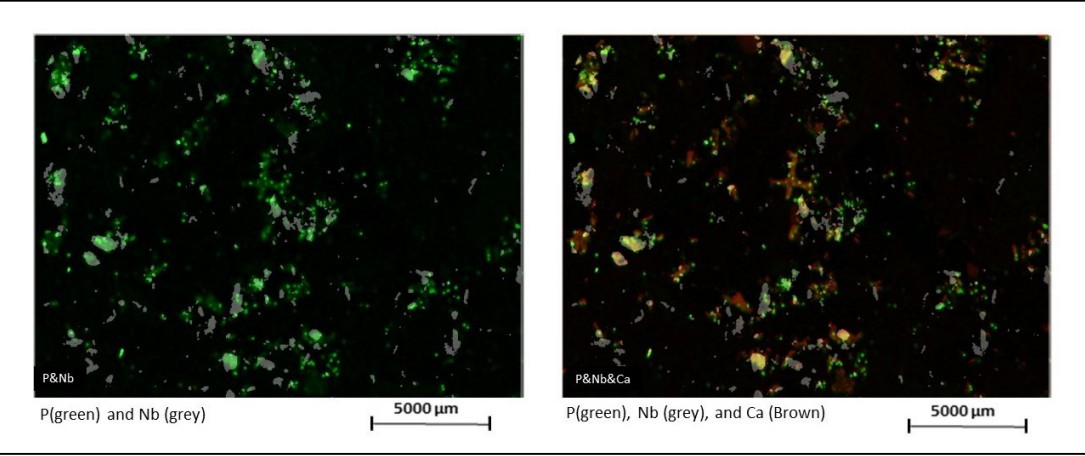

**Figure 14.** Elemental spatial micrometric distributions of P&Ca and Nb&Ca.

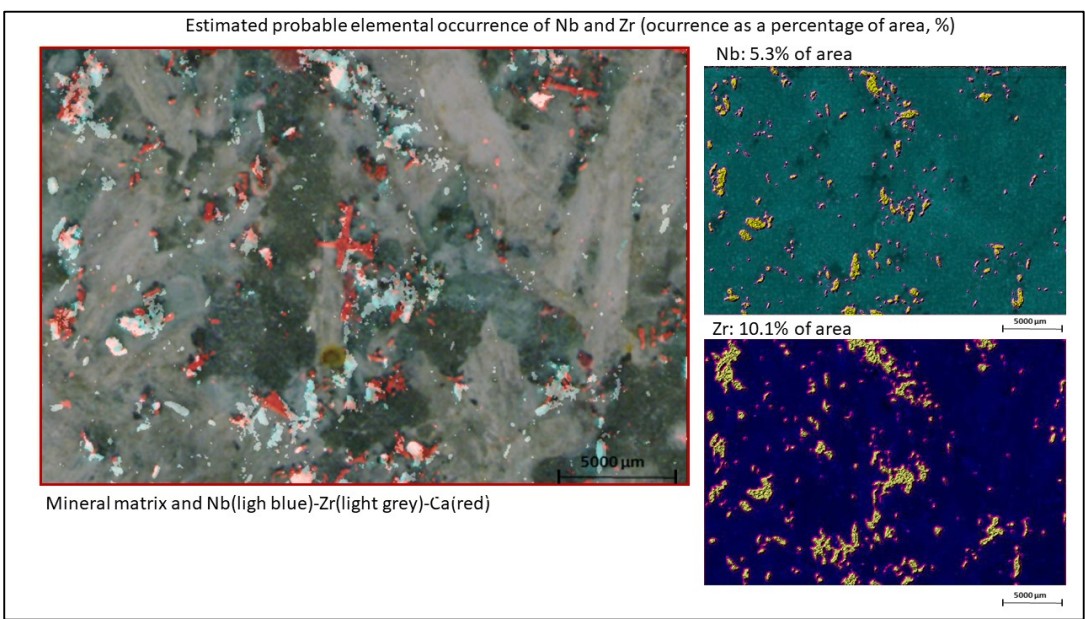

**Figure 15.** Nb, Zr, and Ca spatial micrometric distributions in mineral matrix; Nb and Zr estimated occurrence as a percentage of area (%) through the method "Two colors—spheric search radius criteria" as described in Section 2.3.

### 3.1.1. K, Al, Na, and Cl Silicate Phase Minerals

In Figure 2, µ-EDXRF maps are presented for Si (Figure 2b), K (Figure 2c), Al (Figure 2d), Na (Figure 2f), and Cl (Figure 2g). The estimated percentage of elemental area of occurrence for K and areas without K are also presented in the interpretative K map (Figure 2e). The existence is clear of a mineral matrix almost totally enriched in Si, Al, and K. Si (light to dark green areas in Si map) indicates silicate minerals. In the Si map, the black areas mark minerals such as oxide, disseminated sulphide, and phosphate without any Si in their composition. The K map marks the presence of nepheline minerals (dark green), $Na_3K(Al_4Si_4O_{16})$, and alkali-feldspar (K-feldspar), $KAlSi_3O_8$ (light to medium green). The effects of bladed, Carlsbad-twinned crystals of K-feldspar are clear due to some differences in green color intensities in the K map. These differences in green color intensities can also be explained by the presence of impurities and inclusions in K-feldspar minerals. The estimated area of occurrence of K is indicated in two distinct pink colors and dark green color in the interpretive K map. The estimated area with K is about 85.55%, and it is

mainly related with K-feldspar (pink colors, 55.42%), and nepheline (dark green, 30.13%). The estimated area without K is about 14.45% (in yellow). Most areas of the Na map are marking nepheline, Na-feldspar $NaAlSi_3O_8$, and sodalite $Na_4Al_3(SiO_4)_3Cl$. Sodalite is also marked by most areas of the Cl map. Therefore, apart from Si, Al, and Na, Cl has a low spatial correspondence with the other studied elements. This is the example of P whose map indicates the locations of phosphate mineral apatite (formula of the admixture most common elements: $Ca_{10}(PO_4)_6(OH,F,Cl)_2$) and to which a good or sufficient correspondence with Cl is to be expected. However, as can be observed in Figure 3 concomitant localizations between Cl and P (Cl&P) are almost absent or inexistent which could indicate the depletion effect of Cl in apatite minerals probably due to a genetic effect in the original magmatic ascension process. Distinct authors refer to these phenomena in their studies as a complex chemical zoning in plutonic apatite grains, including effects of F enrichment and Cl depletion [41,42].

In NS, two groups of incompatible elements that have difficulty entering the solid phase are typically present in the mineral matrix [5]: (1) large-ion lithophile (LILE), with large ionic radii, such as K, Rb, Cs, Sr, and Ba, and (2) high-field-strength elements (HFSEs), which include elements of large ionic valences such as Zr, Nb, Hf, REEs, Th, U, and Ta. In Figures 4 and 5 it is possible to observe the high spatial correspondence between the elements Si, Al, K, Rb, Sr, Ga, and Ba. It seems, therefore, that Rb, Ga, Sr, and Ba follow the silicate, and especially aluminium tectosilicate in the mineral matrix of NS. This correspondence is mainly due to the existence of close relationships between ionic radii and high charge ratio (radius/valence) of these elements with K which facilitates its incorporation in aluminium tectosilicate minerals, such as feldspars and nepheline. For, instance, it is known that Rb (Figure 4) is very related to K due to its chemical similarities [43,44] which enables Rb to be substituted for K in alkali-feldspar mineral structures during the process of crystallization. In igneous minerals, Sr is compatible in feldspar having the exact value of the mineral/melt distribution coefficients varying with feldspar composition [45]. This can explain the high and exceptional correspondence between K and Sr detected at micrometric scale (Figure 4). Ga is as chalcophile element and is present in NS generally in interesting concentrations. Usually, Ga (0.62 Å) is completely camouflaged in minerals of aluminium (0.57 Å) and does not form minerals of its own [46]. This explains the very well distributed correspondence of Ga with Si, K, and specially with Al (Figure 4) in the aluminium tectosilicate (nepheline and k-feldspar).

In addition, some of the referred incompatible elements present in NS can be related to the substitution of K in the later stages of crystallisation, which corresponds to magmas of more felsic/acid compositions [44,47], or to weathering. This is the case with $Ba^{2+}$ ion which occurs mostly in K-feldspar and mica and can be enriched by late weathering of magmatic rocks [44,47]. In [48], the authors conclude that the specific mineral phases carrying Sr-Ba are late and formed at low temperatures in oxidizing acidic environments. In the mineral matrix of the analyzed NS, Ba is widely distributed not only through felsic but also ferromagnesian and oxide minerals. It is to say that Ba concentrations tend to be higher in K-feldspars and biotite [46,48] but the $Ba^{2+}$ ion can also substitute for $Ca^{2+}$ in plagioclase, pyroxenes, and amphibole, and in the non-silicate minerals apatite and calcite [47]. Ba can also be adsorbed in the surface of hydrous Fe and Mn oxides, such as $MnO_2$ and $TiO_2$ [47]. The versatility of Ba geochemical behavior may explain the Ba dispersion obtained in μ-EDXRF map distribution (Figure 5).

### 3.1.2. Mg, Ca, and Transition Metals

In Figures 6 and 7 results are presented for Zn and Ni, respectively. Zn and Ni belong to the first-row transition series of elements. As it can be seen, these two transition scarce metals are distributed along the mineral matrix especially in silicate-phase minerals. In Figure 6d,e, correlation between Zn&Fe (Zn and K, Figure 6d), and Zn&S (Zn and K, Figure 6e) S, marking the potential occurrence of disseminated sphalerite ZnS, is very low and comparatively restricted to correlation between the elements Zn&K (Zn and K,

Figure 6f) or Zn&Al (Figure 6g). In magmatic processes zinc acts as chalcophile metallic element. It forms several types of minerals including sphalerite ZnS, smithsonite $ZnCO_3$, and zincite ZnO. However, Zn can also be present as a trace element substitution in other accessory minerals—mainly pyroxene, amphibole, mica, garnet, and magnetite. In this last case, and especially in the case of alkaline complexes, Zn behaves as a lithophile element substituting for $Mn^{2+}$, $Fe^{2+}$, $Fe^{3+}$, and $Al^{3+}$ in six-coordination, being readily partitioned into oxide and silicate minerals by substitution for $Fe^{2+}$ and $Mg^{2+}$, both of which have similar ionic radii to $Zn^{2+}$ (74 pm) [49,50].

Ni occurs widely as both elemental and divalent cationic species, substituting for Fe and Mg in common silicate structures and forming Fe/Ni metal alloys [51,52]. In nature, Ni tends to be present in significant concentrations in ultramafic to mafic formations and in its respective weathering products. Recent μ-XRF studies allow us to conclude that, in residual soil from mafic to ultramafic formations, Ni is often correlated with Fe and Mn at a micrometric scale, and generally does not correlate with Cr, Zn, Ca, or K. However, this is not the case with the results of this study. In fact, Ni is widely distributed along the silicate mineral matrix having very good correlations with K (Figure 7d), Al (Figure 7e), and, inclusively, with Zn (Figure 7f). As in the case of Zn, the correlation between Ni and Fe (Figure 7g) is not very evident, being restricted to some minor locations with low, very micrometric dimensions. These common behaviors of Zn and Ni lead to the possibility that other transition elements may be preferably present in silicate minerals due to substitution effects having a low representation as sulphide, sulphate, or oxide. In our specific case study, other transition elements that also belong to the first-row have, however, a distinct behavior comparatively to Zn, and Ni. This is the case with the elements Fe, Ti, Mn, and Co (Figures 8 and 9).

Figure 8 presents interpretative maps where areas with total absence of Si (Figure 8b) and Al (Figure 8c) are presented. As can also be seen from Figure 8, Fe occurs in a mineral matrix in areas partially without Si and with complete absence of Al. This is also observed for elements Mg, Ca, and Co (Figure 9). This highlights the antagonism distribution of Al and Si (partially) relatively to Fe, Ca, Mg, Mn, and Co. Fe, Ca, and Mg, can work as indicators of ferromagnesian minerals such as pyroxenes, amphiboles, and micas. In the studied sample, they can be marking minerals such as biotite, aegirine-augite, and hornblende [28,29]. Oxides such as Fe-oxide magnetite are present in Monchique NS [28,29]. An interesting aspect is the correspondence of Co with Fe, Ca, Mg, and Mn. Co is strongly coherent with Mg in igneous rocks and behaves like magnesium. In [29] the author observed that transition elements, such as Co, Cr, V, and Sc are predominantly incorporated in mafic phases of Monchique NS such as olivine, clinopyroxene, amphibole, and magnetite. Furthermore, in sediments, most of the Co is in the argillaceous fraction and seems to follow iron and manganese [53]. The ionic radii of $Co^{2+}$ and $Co^{3+}$ are similar to the ionic radii of $Mg^{2+}$, $Mn^{4+}$, $Fe^{2+}$, $Fe^{3+}$, and $Ni^{2+}$. Co can substitute for any of these elements in many minerals [54].

### 3.1.3. Ti, HFSE, and Zr-REE-Nb Enrichments

Syenite magmatic rocks are characterized by a remarkable enrichment of HFSEs, including Zr, Nb, Y, and U as well as REEs [4]. Following these possibilities μ-EDXRF maps of elements Ti, Fe, Zr, Nb, Ca, and P were analysed. As can be observed in Figure 10, it is clear that there is a very weak spatial correspondence between Ti and Fe, which have, in fact, an antagonistic behavior occurrence. Alternatively, there is a very good spatial correlation between Ti and Ca. These two facts lead to the assumptions that (1) Ti is present as sphene $CaTiSiO_5$, and as Ti-oxide $TiO_2$ (rutile), in this last case, in locations with absent Si, and (2) there is a very low probability of Ti to be present as a Fe-Ti oxide $FeTiO_3$ (ilmenite).

Figure 11 shows the spatial relationships between Zr&Si, and in turn, Zr&Si&Ca. Detailed observation of these two micrometric maps allows us to confirm the existence of Zr as zircon $ZrSiO_4$ or $(Zr_{1-y}, REE_y)(SiO_4)_{1-x}(OH)_{4x-y}$, or even, as the oxide zirconolite—$CaZrTi_2O_7$ or $(Ca,Ce)Zr(Ti,Nb,Fe^{3+})_2O_7$.

Nb and Zr have a very good spatial correspondence at micrometric scale (Figures 12e and 13) which can be corroborated with the spatial correlation coefficients PCC (values between 0.84 and 0.91), and MOC (values from 0.93 to 0.98) (Figure 13). In turn, Fe has a very low correspondence with Ti (Figure 12c) and Nb (Figure 12f). In addition, the spatial correlation of Ti and Zr is not so obvious although it is possible to identify some superpositions (Figure 12b). These results may be indicative that Nb will have replaced Ti in sphene or Zr-oxide, highlighting the possibility of existence of Zr-REE-Nb enrichments. In NS, Ti and Zr occur mainly as oxides within the mineral matrix or as impurities in some minerals such as apatite $Ca_5(PO_4)_3(OH, F, Cl)$ [12,13]. Nb generally occurs commonly as a substituent of Ti and replacement effects with enrichments of REEs [4,13] are common in some specific minerals such as apatite $Ca_5(PO_4)_3(OH, F, Cl)$, zircon, $ZrSiO_4$ or $(Zr_{1-y}, REE_y)(SiO_4)_{1-x}(OH)_{4x-y}$, and zirconolite, $CaZrTi_2O_7$ or $(Ca,Ce)Zr(Ti,Nb,Fe^{3+})_2O_7$ (when replacements occurs). In Figure 14 it is possible to observe the spatial relationships between the elements P&Nb and P&Nb&Ca. The good spatial co-existence between P&Nb and P&Nb&Ca also enhances the possibility of existence of Nb in apatite minerals. Figure 15 shows the superposed distributions of Zr, Nb, and Ca in the mineral matrix sample. The elemental occurrences of Nb and Zr (estimated probable elemental occurrence as a percentage of area (%) are also shown. The proportion between Nb and Zr is $\frac{1}{2}$ which seems adequate to indicate the possibility of Nb being a substitute HFSE.

## 4. Discussion and Conclusions

The results found for the NS of "Serra de Monchique" are quite promising since they may indicate the existence of significant enrichments of Ti, Zr, and Zr-REE-Nb in individualized mineral phases such as oxides, zircon, Zr-oxides, sphene, and apatite which can be more easily separated in the mineral beneficiation process. Nb and light-REE concentrations referenced in [29] evidence this interesting possibility of recovering some CRM from Monchique NS. It can be considered that Y, and probably Hf, which could not be detected with μ-XRF surveys, will also be present in the Monchique NS mineral matrix.

A key issue in the possibilities of processing NS is related with the remarkable potential for secondary mineral valuation of different CRMs which are present in the impurities to be separated from feldspathic minerals. Possibilities on processing NS under this objective are referenced by distinct authors and are being increasingly tested by several mining operators. Research into the purification of raw materials by removing accessory minerals found as inclusions within feldspar, such as apatite and rutile, or secondary minerals formed as a result of alteration and/or weathering have been performed for many years [4,6,9,11]. These processes include handwashing, breaking, grinding, classifying according to size, magnetic or electrostatic separation, and flotation [11]. Magnetic separation partitions magnetic (iron-rich) minerals, including those carrying REEs, from non-magnetic minerals and is quite an interesting possibility [11]. Spiral classification associated with flotation and magnetic separation can be very effective in separating distinct silicate phases that include iron impurities, such as mica, from feldspar minerals [11]). Investigation of microwave roasting for potash extraction from NS is also discussed in [55]. In [6] the authors refer to flotation and high gradient magnetic separation (HGM) as beneficiation processes to be used to process REE-bearing minerals such as zircon.

However, igneous rocks such as NS display a remarkable spatial variability and heterogeneity in terms of mineral composition, grain size, mineral association, and texture which leads to low liberation sizes of ore or accessory minerals-of-interest reducing efficiency and sustainability of the beneficiation process [6]. The situation is even more complicated and inefficient when, in addition, the processing of fine fractions comprises phases with different physical properties (e.g., density, magnetic susceptibility) [6]. Such disadvantages for efficient, economic, and sustainable mine-processing represent a barrier to CRM recovery possibilities associated with alkaline granites and syenites. Future geometallurgical characterization studies will be necessary to better identify mineral textures and establish the liberation size of the distinct recognized potential resources. It is therefore necessary to

promote detailed mineralogical studies to select the most appropriate and economic separation techniques [6,11]. Different authors have evidenced the potentials of the application of μ-XRF and scanning electron microscopy (SEM) techniques, especially when their distinct quantitative results are combined for improved interpretations [14–16]. Other spectroscopic methods such as micro-Raman and mineral liberation analyzer (MLA) techniques represent promising ways to improve knowledge in this context.

Processing technologies for REE-bearing silicates require additional research and development (R&D) investment to make them commercially viable [56,57]. The main problems to be solved are related with fine-grain liberation size and interference of impurities in the metallurgical concentration process. In general, comminution to low grain size is an energy- and material-consuming step, belonging to the most expensive positions in processing operations. Low liberation grain sizes also constrain processing activities, mainly leaving flotation as the most possible suitable technique enhancing operation costs, and water and supplies consumption [4,56,58,59]. Recent developments in individual REE separation technologies in both metallurgical and recycling operations have been highlighted by [57]. Depending on mineralogy and reactivity of gangue phases, the extraction of REEs usually involves roasting of the ore and dissolution of the ore using acidic or alkaline solutions. Generally, separation techniques such as solvent extraction, ion exchange, and precipitation are used for the recovery of REEs from pregnant leach solutions (PLS) obtained from acid or alkali leaching [57]. As referenced by [57], in recent times, a revolutionary technology called SuperLig® molecular recognition technology (MRT) has been increasingly used to selectively separate and recover individual REEs [60]. Potential applications include REE recovery from primary ore, tailings, coal ash, and spent industrial feedstock such as permanent magnets, rechargeable batteries and LED lighting systems [61]. Combination of small REE-bearing mineral particle size and complex REE mineralogy in coal-based resources has also led to intensive investigation for possible and feasible REE recovery techniques [57,62]. Physical beneficiation, acid leaching, ion-exchange leaching, bio-leaching, thermal treatment, alkali treatment, solvent extraction, and other recovery technologies have been evaluated with varying degrees of success depending on the feedstock properties. In general, physical beneficiation can be a suitable low-cost option for preliminary upgrading. However, most studies showed exceedingly low recovery values unless ultrafine grinding was first performed [62]. In [62] the authors concluded that direct chemical extraction by acid was able to produce moderate recovery values, and the inclusion of leaching additives, alkaline pre-treatment, and/or thermal pre-treatment considerably improved the process performance.

**Author Contributions:** Conceptualization, S.B. and A.D.; methodology, S.B. and A.D.; software, S.B., A.D., D.D., J.G. and G.B.; validation, S.B., A.D., S.P. and J.S.; investigation, S.B., A.D., D.D., J.G., G.B. and J.C.; writing—original draft preparation, S.B.; writing—review and editing, G.B., A.D., S.P., J.S. and J.A. All authors have read and agreed to the published version of the manuscript.

**Funding:** This research was funded by FCT-Fundação para a Ciência e a Tecnologia, Portugal, grants number UIDB/04035/2020, and UID/FIS/04559/2020.

**Data Availability Statement:** Not applicable.

**Acknowledgments:** The authors acknowledge the support of LIBPhys, GeoBiotec, Department of Physics and Department of Earth Sciences of Nova School of Science and Technology for the development of the laboratory work.

**Conflicts of Interest:** The authors declare no conflict of interest.

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
