# Peer review of "Exploring High-Resolution Chemical Distribution Maps of Incompatible and Scarce Metals in a Nepheline Syenite from the Massif of “Serra de Monchique” (Portugal, Iberian Peninsula)"

_minerals, doi:10.3390/min12091178_

Round 1
Reviewer 1 Report
The paper is of great scientific and economic interests and I think it fits the journal’s scope. I provided some corrections and suggestions that I think could be useful to improve the paper. Abundant Zr-, REE- and Nb-bearing independent minerals, such as columbite group minerals, samarskite, pyrochlore, bastnaesite and so on, usually occur in nepheline syenite. Obviously, SEM is a more efficient technique to identify these minerals in NS. Adding some SEM data would be very helpful. Moreover, I am worried about the representation of the sample. Only a small area from one sample was selected for μ- EDXRF surveys.
Specific comments
L115: It is should be “The nepheline syenite of “Serra de Monchique” Massif
L129: Please mark the main minerals (K-feldspars, nepheline, aegirine-augite) in Figure 1c.
L122: Whether the Monchique massif show any petrographical zoning. Please introduce more.
L144-145: You do not mention whether your rocks are agpaitic or miaskitic although you introduced the nomenclature here. Moreover, By the original definition requiring (Na+K)/Al >1.2 (Ussing, 1912), the rock is not agpaitic. In modern petrographic nomenclature, the definition of agpaicity is related to the presence of critical HFSE-bearing minerals (e.g. eudialyte group minerals, Na-Ca-Zr-Ti-F disilicate minerals) instead of zircon, titanite and ilmenite that are characteristic of miaskitic nepheline syenite (Le Maitre, 2003, following Sørensen, 1960).
L284: There should be a space blank between the words “it” and “is”.
L290-294:In Figure 4, it seems that Sr and K have a good correspondence. However, Rb seems to have a better correspondence with Si or Al rather than K.
L320: “As it can observed” should be “As it can be observed”.
Figure 6: What is the difference between Fig. 6b and Fig. 6c.
L328-331: I think Zn is very related with Fe and Mn. So, could you put the correspondence of Zn with Fe and Mn in Figure 6.
L332-345: Like Zn, it is better to put the correspondence of Ni with Fe and Mn in Figure 7.
L375: Please replace “where” with “were”.
L389: I wonder how about the spatial correspondence between Nb and Fe.
L442: As far as I know, most REE deposit associated with alkaline rocks/ alkaline granites could not be mined due to their tiny grain size. Please introduce recent information about this.
Author Response
Dear Reviewers,
Thank you very much for your consideration, contributions, and comments.
The manuscript has been reviewed following your comments, corrections, and suggestions (detailed response list in the attached file). The changes that have been applied are listed and described below and include: minor revisions of typing errors in the text, improvements in figures 1, 4, 6, 7, and 12, introduction of some new texts, and respective references for better support.
We hope that these revisions are accordingly with your expectations, and we thank you for your detailed revision work which has significantly improved the quality of our article.
Thank you in advance,
Sofia Barbosa, corresponding author

Reviewer 2 Report
Dear authors,
my main concerns are the following:
(1) the procedure of sample polishing should be described in detail since it can affect the sample chemistry (for example, the use of pastes of different composition);
(2) since the article is more about the method, I did not have enough description of the specifics of these results in the Discussion. Are they the same as obtained by other methods? Are they better? If yes, then what? How long does such an analysis take?
For example, I am a user of SEM EDS where such maps can be obtained in a couple of hours from polished sample. In which cases and why should I use 2D micro Energy Dispersive X-ray Fluorescence?? I do not want direct answer to my question in your paper (sure:)), but many authors will end up with the same questions.
Minor comments are marked in pdf.

Author Response
Dear Editors,Dear Reviewers,
Thank you very much for your consideration, contributions, and comments.
The manuscript has been reviewed following your comments, corrections, and suggestions (detailed response list in the attached file) The changes that have been applied are listed and described below and include: minor revisions of typing errors in the text, improvements in figures 1, 4, 6, 7, and 12, introduction of some new texts, and respective references for better support.
We hope that these revisions are accordingly with your expectations, and we thank you for your detailed revision work which has significantly improved the quality of our article.
Thank you in advance,
Sofia Barbosa, corresponding author
